# The Impact of Restricted versus Liberal Early Fluid Volumes on Plasma Sodium, Weight Change, and Short-Term Outcomes in Extremely Preterm Infants

**DOI:** 10.3390/nu14040795

**Published:** 2022-02-14

**Authors:** Barbro Diderholm, Erik Normann, Fredrik Ahlsson, Richard Sindelar, Johan Ågren

**Affiliations:** Department of Women’s and Children’s Health, Uppsala University Children’s Hospital, SE-75185 Uppsala, Sweden; erik.normann@kbh.uu.se (E.N.); fredrik.ahlsson@kbh.uu.se (F.A.); richard.sindelar@kbh.uu.se (R.S.); johan.agren@kbh.uu.se (J.Å.)

**Keywords:** extremely preterm infant, fluid allowance, dehydration

## Abstract

The optimal fluid requirements for extremely preterm infants are not fully known. We examined retrospectively the fluid intakes during the first week of life in two cohorts of extremely preterm infants born at 22–26 weeks of gestation before (*n* = 63) and after a change from a restrictive to a more liberal (*n* = 112) fluid volume allowance to improve nutrient provision. The cohorts were similar in gestational age and birth weight, but antenatal steroid exposure was more frequent in the second era. Although fluid management resulted in a cumulative difference in the total fluid intake over the first week of 87 mL/kg (*p* < 0.001), this was not reflected in a mean weight loss (14 ± 5% at a postnatal age of 4 days in both groups) or mean peak plasma sodium (142 ± 5 and 143 ± 5 mmol/L in the restrictive and liberal groups, respectively). The incidences of hypernatremia (>145 and >150 mmol/L), PDA ligation, bronchopulmonary dysplasia, and IVH were also similar. We conclude that in this cohort of extremely preterm infants a more liberal vs. a restricted fluid allowance during the first week had no clinically important influence on early changes in body weight, sodium homeostasis, or hospital morbidities.

## 1. Introduction

The evidence base to guide initial fluid management for extremely preterm infants is limited. A few randomized controlled trials [1,2,3,4,5] have studied the impact of different levels of either sodium, water, or both sodium and water provision, on survival and the risk of cardiorespiratory morbidity. The regimes applied in these investigations were different, but in those where sodium intake was varied separately from water a restricted sodium intake resulted in a lower rate of bronchopulmonary dysplasia (BPD) and a lower mortality. Based on these studies the most current recommendations [6] suggest an initial restriction of both water and sodium intake, presumably since in most instances an increased total fluid volume will also infer an increased amount of sodium provided. Importantly, these studies include a limited number of infants with a gestational age (GA) below 28 weeks and might thus not provide much guidance to the care provided to this subpopulation. The aim of the present investigation was to study the effects of an implemented change in the initial fluid volume provision on early measures of hydration and hospital outcomes in a cohort of extremely preterm infants. The change to higher starting fluid volumes was initiated with the ambition to improve energy and nutrient provision and was not primarily driven by perceived difficulties with excessive weight loss and/or electrolyte imbalances.

## 2. Materials and Methods

The investigation evaluated an implemented increase in the total starting fluid volume allowance to extremely preterm infants at a tertiary care neonatal intensive care unit (Uppsala University Children’s Hospital).

### 2.1. Subjects

Data were retrospectively collected from all in-born (*n* = 204) extremely preterm infants with a gestational age (GA) of 22–26 weeks, during a 6-year period (2002–2007). By design this included a 3-year period of lower fluid volumes (RESTRICTED) before the change in fluid allowance, and a 3-year period after the implementation of the higher fluid volume allowance (LIBERAL). Infants who died or were transferred to other units within the 1st week were excluded, leaving a total of 175 infants for further analysis (RESTRICTED *n* = 63; LIBERAL *n* = 112) with a mean GA of 25^1/7^ (range 22^1/7^–26^6/7^) weeks (Figure 1).

The groups were similar in GA and birth weight (Table 1). The administration of antenatal steroids <24 h prior to delivery was more frequent in the later (LIBERAL) era (*p* < 0.05, Table 1). Including all infants admitted, survival to discharge was 77/102 (75%) and 119/140 (85%) in RESTRICTED and LIBERAL eras, respectively (*p* = 0.0696).

### 2.2. Study Setting

All infants were cared for in closed intensive care incubators set at a high relative humidity of 85% during the first week, after which a setting of 50% was used. In brief, the institutional fluid prescription guidelines specified a starting total fluid volume and a standard flush solution (NaCl 9 mg/mL at 0.5 mL/h) for umbilical arterial lines. Enteral feeds (donor breast milk, later replaced with Mom’s own) were to be started within 2 h with a volume of approximately 20 mL/kg/d and with a recommended increase of approximately 20 mL/kg/d until full feeds (170 mL/kg/d) were reached. Parenteral nutrition (PN; amino acids, glucose, and lipids) were also started at birth. Supplementation of sodium was not to be provided until weight loss and a drop in serum sodium had occurred. In the RESTRICTED group an initial total fluid volume of 65 mL/kg/day was recommended with a scheduled daily stepwise increase of 10 mL/kg/day up to 170 mL/kg/day. In the LIBERAL group total fluids were started at 95 mL/kg/day (GA 22–24 weeks) and 85 mL/kg/d (GA 25–26 weeks), respectively. In the LIBERAL era the recommended daily increase in fluid volume allowance was also 10 mL/kg/d to be adjusted at the discretion of the attending neonatologist, depending on changes in body weight (determined daily), urinary output, blood biochemistry values (determined q 6 h until stable), and clinical status. The LIBERAL era guidelines advocated for fluid management also to be guided mainly by the serial determinations of body weight to allow a weight loss of up to 3–4% per day and a total maximum weight loss of 10–15%. During the study period a subset of infants (*n* = 41) born after October 2005 received an amino acid flush solution in their umbilical artery catheter [7], but no other changes were made in the guidelines for enteral or parenteral nutrition or for transfusions.

### 2.3. Data Collection and Treatment

Data were collected from the patients’ records and fluid charts and included details on all fluids provided (volume and route of administration), transfusion volumes, postnatal changes in body weight, and blood biochemistry values. Hypernatremia was categorized as moderate (P-Na > 145 mmol/L) or severe (P-Na > 150 mmol/L), and hyponatremia was defined as a P-Na of <130 mmol/L. Further, data on selected short-term outcomes considered relevant to early fluid provision were collected; including the incidence of intraventricular hemorrhage (IVH), surgical ligation of persistent ductus arteriosus (PDA), and bronchopulmonary dysplasia (BPD) (defined as oxygen dependency at 36 weeks postmenstrual age).

The values are presented as mean ± SD or median (range), the Student’s *t*-test on unpaired observations and the Fisher’s exact test were used to test for statistical significance, and *p* < 0.05 was considered statistically significant.

## 3. Results

The starting fluid volume intake was 19 mL/kg/d higher (*p* < 0.001) in LIBERAL vs. RESTRICTED (Table 2) and subsequently resulted in an accumulated fluid volume excess of 87 mL/kg for the 1st week (*p* < 0.001, Table 2).

No differences were found in the accumulated volume of transfusions or in the feeding strategy, as demonstrated by the ratio of enteral vs. parenteral nutrition in the groups (Table 2). Thus, the higher total fluids in LIBERAL consisted of higher volumes of parenteral nutrition. Irrespective of the distinct differences in fluid provision, the two groups displayed a similar peak in plasma sodium concentration at a postnatal age of 3 days (Table 2). The trajectories of P-Na are displayed in Figure 2.

Most interestingly, the groups were also similar and parallel in their postnatal changes in body weight (Figure 3), including a nadir at a postnatal age of 4–5 days (Table 2). Further, there were no differences between the two groups in the incidence of hyper- and hyponatremia (Table 3) and the groups had a similar incidence of IVH, PDA, and BPD (Table 3).

## 4. Discussion

The present investigation presents data from a cohort of two 3-year epochs of extremely preterm (GA 22–26 weeks) infants before and after an implemented change in the total fluid volume allowance during the first week of life. The two different levels of fluid volume provided had no discernible impact on weight loss, plasma sodium (P-Na), or the incidence of either hyper- or hyponatremia. In addition, there were no significant differences in the incidences of the selected hospital outcomes between the groups.

The results are consistent with several randomized trials with a large variation in design and outcome measures, but which have all included preterm infants [8]. Two previous studies that investigated the effect of two different levels of water intake—while sodium intake was equivalent [3,5]—demonstrated no effect on survival and/or the incidence of BPD, while studies where sodium intake varied [1,2,4,9] seem to favor the restriction of early sodium intake. These investigations are indirectly supported by more recent data obtained from infants below 27 weeks in the Swedish EXPRESS study [10]. This investigation demonstrated that there seems to be little impact of fluid volume per se on P-Na, while both P-Na and the incidence of hypernatremia correlate with the amount of sodium provided. Altogether, these findings support the notion that early sodium provision should be restricted until S-Na drops and weight loss is underway [11,12]. Indeed, extremely preterm infants also seem to have the ability to regulate urinary output and thus to handle an interval of fluid volumes in the maintenance range, even early after birth [13,14,15].

In parallel with recently presented data from infants born at 22 and 23 weeks at our institution [16], the incidence of hypernatremia is low compared to what has been published from similar populations [10,17,18,19]. Preliminary data from the EXPRESS cohort supports the notion that this might at least partly be explained by the relatively restrictive transfusion policy applied at our institution (C. Späth et al., manuscript in preparation).

The present investigation is limited by its retrospective nature, and albeit being from a single center with an explicitly guideline-driven fluid management and standardized care environment we cannot exclude that changes in the clinical management over time might have influenced our findings. Regrettably, we do not have access to detailed calculations of sodium intake that might have influenced the findings, and it is conceivable that some infants in the LIBERAL group could have received a higher amount of sodium since the increase in fluids was from parenteral nutrition containing sodium. In addition, the hospital outcomes chosen are at best crude representations of perturbations in fluid balance and/or cardiorespiratory function. Nevertheless, we strongly believe that the cohort and its management to be relevant to the care currently provided to this group of highly vulnerable infants. In this context, it may be reassuring to conclude that there seems to be an interval of fluid volumes that may well support near-optimal fluid homeostasis.

## 5. Conclusions

In a cohort of extremely preterm infants the implementation of a more liberal fluid volume allowance had no influence on early changes in body weight, plasma sodium concentrations, the incidence of hypernatremia, short-term growth, or the occurrence of IVH, PDA ligation, and BPD. As long as early sodium load is limited, the implementation of a higher starting fluid volume allowance to achieve better nutrient/caloric intakes might be a feasible approach.

## Figures and Tables

**Figure 1 nutrients-14-00795-f001:**
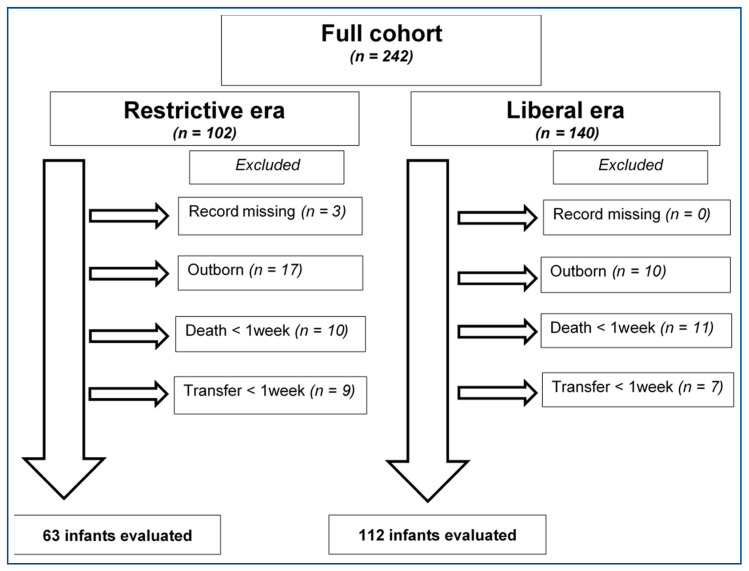
Included and excluded infants in the study cohort.

**Figure 2 nutrients-14-00795-f002:**
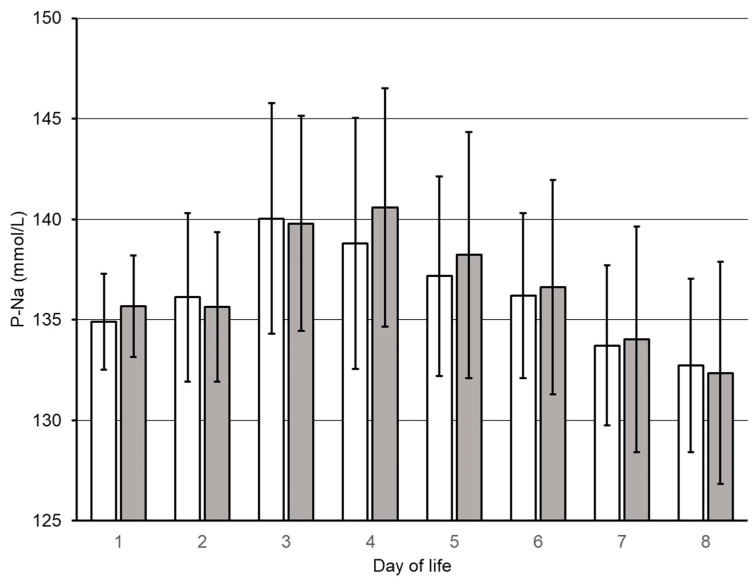
Trajectories of first week plasma sodium (P-Na) concentrations. RESTRICTED group, open bars; LIBERAL group, filled bars; Mean ± SD.

**Figure 3 nutrients-14-00795-f003:**
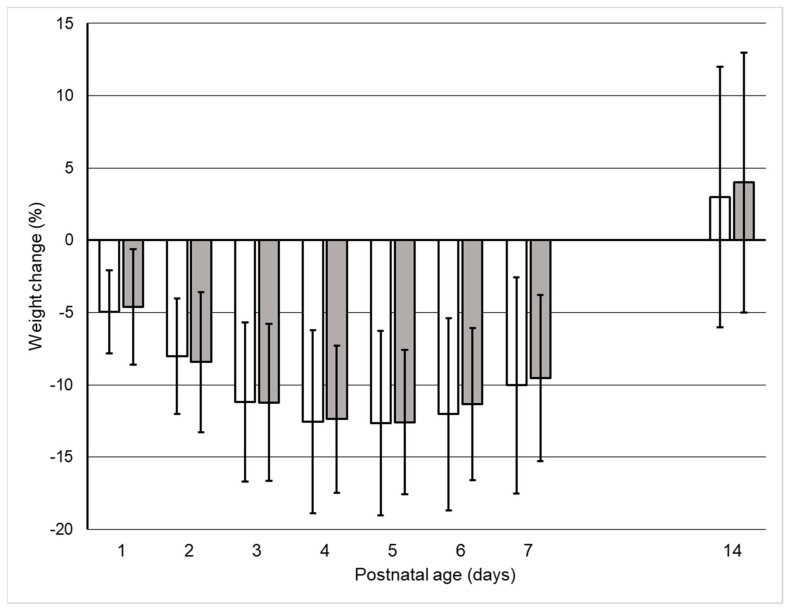
Relative changes in body weight. RESTRICTED group, open bars; LIBERAL group, filled bars; Mean ± SD.

**Table 1 nutrients-14-00795-t001:** Study cohort.

	RESTRICTED (*n* = 63)	LIBERAL (*n* = 112)	*p* ^a^
Gestational age (weeks)	25.2 ± 1.2	25.1 ± 1.1	n.s. ^a^
Birthweight (g)	744 ± 192	718 ± 156	n.s. ^a^
Antenatal steroids	50 (79)	103 (92)	<0.05 ^b^
Surfactant	57 (90)	107 (96)	n.s. ^b^

Values are mean ± SD, or *n* (%). ^a^ Unpaired *t*-test, ^b^ Fisher’s exact test.

**Table 2 nutrients-14-00795-t002:** Fluid volumes, weight change, and plasma sodium.

		RESTRICTED	LIBERAL	
		(*n* = 63)	(*n* = 112)	*p* ^a^
**Total fluids**	(mL/kg/day)			
DOL 1		72 ± 12	91 ± 15	<0.001
DOL 4		110 ± 14	118 ± 14	<0.001
DOL 8		143 ± 20	150 ± 19	0.056
DOL 11		168 ± 15	171 ± 14	n.s.
**1st week**				
Total fluids	(mL/kg/day)	111 ± 11	122 ± 12	<0.001
Transfusion volume	(mL/kg)	15 ± 16	16 ± 14	n.s.
Ratio EN:PN		1:1.7	1:2.1	n.s.
**Weight change**				
Nadir	(%)	−14 ± 5	−14 ± 5	n.s.
Age at nadir	(days)	5 (1–7)	4 (2–7)	n.s.
DOL 8	(%)	−9 ± 7	−10 ± 6	n.s.
DOL 28	(%)	+29 ± 15	+29 ± 12	n.s.
**Plasma sodium**				
Peak	(mmol/L)	142 ± 5	143 ± 5	n.s.
Age at peak	(days)	3 (1–7)	3 (1–7)	n.s.
Sodium supplementation	(%)	50 (80)	90 (80)	n.s.
Age at start	(days)	7 (2–17)	7 (3–13)	n.s.

Values are mean ± SD, median (range), or *n* (%). DOL—Day of life; EN: PN—Enteral: Parenteral nutrition; ^a^ Unpaired *t*-test.

**Table 3 nutrients-14-00795-t003:** Plasma sodium during 1st week and related morbidities.

	RESTRICTED	LIBERAL	
	(*n* = 63)	(*n* = 112)	*p* ^a^
**Hypernatremia**			
>145 mmol/L	18 (29)	40 (36)	n.s.
>150 mmol/L	6 (10)	12 (11)	n.s.
**Hyponatremia**			
<130 mmol/L	13 (21)	24 (21)	n.s.
**Hospital morbidity**			
BPD	24 (38)	41 (36)	n.s.
PDA ligation	20 (32)	39 (34)	n.s.
IVH, all grades	16 (25)	21 (19)	n.s.
IVH, grades 3–4	8 (14)	8 (7)	n.s.

Values are *n* (%). BPD—bronchopulmonary dysplasia; PDA—persistent ductus arteriosus; IVH—intraventricular hemorrhage; ^a^ Fisher’s exact test.

## Data Availability

Raw data supporting the results reported in this publication can be found at: https://www.researchgate.net/profile/Johan-Agren/research (accessed on 8 February 2022).

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
