# Peer review of "The Impact of Restricted versus Liberal Early Fluid Volumes on Plasma Sodium, Weight Change, and Short-Term Outcomes in Extremely Preterm Infants"

_nutrients, 2022, doi:10.3390/nu14040795_

Round 1

Reviewer 1 Report

Diderholm and coworkers provide data from a retrospective analysis in extremely low birth weight infants < 27 weeks on the short-term outcomes before and after the change of fluid management in their level 3 NICU. The topic is of importance, the available literature on the topic is limited and studies were executed more than twenty years ago where the NICU management in general and outcome data were highly different from the today´s situation. The manuscript is well-written and the results are presented in a clear manner. I have some suggestions that can help to further improve the quality of the manuscript:

  • The authors should introduce the foundations for their change in fluid management in the abstract and the intro as the reader is left wondering about the strategy that is in contrast to i.e. the Cochrane conclusions in reference 6.
  • The authors state a six year study period, addition of dates and years will precise the period.
  • It will be of interest to the reader if the authors could add some more details on their NICU management including the interval between determination of weight and blood values, the duration of restrictive fluid management or when the steady-state situation with the recommendations according to ESPGHAN were reached. Particularly, the regime of amounts of routine sodium chloride supply within the first days of life would be of interest as the team has no problems with hypernatremia.
  • The authors could provide more details on the respiratory course and management, including duration of MV, NIV and oxygen provision and further details on PDA presence and its pharmacological therapy and postnatal steroids if these data are available.
  • In this context, they should discuss and risk-adjust the disparity in antenatal steroids between the two observation periods. There might be a trade-off effect from increased antenatal steroid exposure to compensate for the more liberal fluid supply that should be stated.
  • The authors provide ratios of enteral and parenteral nutrition. It would be helpful to see the total caloric intake in both groups if available.
  • There are several actual publications from 2021 on the topic that detail variances in fluid, sodium chloride and total nutritional supply in comparable retrospective studies, for example Soullane et al, Pediatr Res 2021. It would be worth to include them into the discussion.

Reviewer 2 Report

This topical focused analysis on fluid management in this specific subpopulation of extreme preterm neonates is valuable and add to the limited literature on this topic. 

I therefore value the efforts of the authors and only can come up with some specific questions to better understand the setting as described. 

The paper clearly mentions the care in closed incubators, with the use of high humidity, but was there also a ‘structured’ weaning strategy on this relatively humidity, starting at 85 % ?

Is it correct that both regimens targeted for the same 170 ml/kg/day ?

Was sodium quantification done daily, or at hoc ? structured ?

Are there any data on diuresis (cfr your claim on ability to regulate urinary output ?

Was caffeine used (serves as diuretic, cfr CAP trial)

Some more additional information on the nutritional management is valuable.

To what extent has this study affected your practices in your unit , how do you interpret the data ?

Any data on delayed cord clamping practices ?

Any data on inotropics needed ?

Round 2

Reviewer 1 Report

The authors have adequately addressed all my points.